# Miltefosine and Nifuratel Combination: A Promising Therapy for the Treatment of *Leishmania donovani* Visceral Leishmaniasis

**DOI:** 10.3390/ijms24021635

**Published:** 2023-01-13

**Authors:** Estela Melcon-Fernandez, Giulio Galli, Carlos García-Estrada, Rafael Balaña-Fouce, Rosa M. Reguera, Yolanda Pérez-Pertejo

**Affiliations:** Department Ciencias Biomédicas, Universidad de León, Campus de Vegazana s/n P.C, 24071 León, Spain

**Keywords:** Leishmania, drug combinations, nifuratel, miltefosine, drug repurposing

## Abstract

Visceral leishmaniasis is a neglected vector-borne tropical disease caused by *Leishmania donovani* and *Leishmania infantum* that is endemic not only in East African countries, but also in Asia, regions of South America and the Mediterranean Basin. For the pharmacological control of this disease, there is a limited number of old and, in general, poorly adherent drugs, with a multitude of adverse effects and low oral bioavailability, which favor the emergence of resistant pathogens. Pentavalent antimonials are the first-line drugs, but due to their misuse, resistant *Leishmania* strains have emerged worldwide. Although these drugs have saved many lives, it is recommended to reduce their use as much as possible and replace them with novel and more friendly drugs. From a commercial collection of anti-infective drugs, we have recently identified nifuratel—a nitrofurantoin used against vaginal infections—as a promising repurposing drug against a mouse model of visceral leishmaniasis. In the present work, we have tested combinations of miltefosine—the only oral drug currently used against leishmaniasis—with nifuratel in different proportions, both in axenic amastigotes from bone marrow and in intracellular amastigotes from infected Balb/c mouse spleen macrophages, finding a potent synergy in both cases. In vivo evaluation of oral miltefosine/nifuratel combinations using a bioimaging platform has revealed the potential of these combinations for the treatment of this disease.

## 1. Introduction

Visceral leishmaniasis (VL) is classified by WHO within the catalog of Neglected Tropical Diseases (NTDs) as a lethal disease caused by certain species of protozoan parasites of the genus *Leishmania* that mainly affects the poorest populations in developing countries [1]. VL is a vector-borne disease that causes severe systemic infections when the pathogenic form of the parasite (amastigote) invades internal organs such as the thymus, liver, spleen and bone marrow, ultimately producing renal dysfunction that can be fatal when patients do not receive adequate drug treatment [2,3]. The current incidence of VL is 50,000 to 90,000 new cases each year (https://www.who.int/es/news-room/fact-sheets/detail/leishmaniasis, accessed on 9 January 2023), with more than 5700 deaths in 2019 according to reports from the Drugs for Neglected Diseases initiative (DNDi) (https://dndi.org/diseases/visceral-leishmaniasis/facts/, accessed on 9 January 2023). The lack of a specific vaccine for the human disease makes therapy dependent on a limited number of obsolete, toxic or poorly absorbable oral drugs that lose efficacy over time due to the emergence of resistant strains [4,5]. Pentavalent antimonials (Sb^V^), namely Glucantime and Pentostam, remain the first-line treatments for different types of leishmaniasis in many endemic countries despite having significant side effects, such as the intrinsic toxicity of antimony [6], the long and painful pattern of intramuscular injections, and the emergence of acquired resistance due to their misuse [7]. Amphotericin B deoxycholate (AMB) and the liposomal formulation of AMB (AmBisome) are highly effective antileishmanial drugs, but must be administered intravenously due to their poor oral bioavailability [8,9]. In addition, they are chemically unstable compounds at the high temperatures of endemic countries, thereby requiring a cold supply chain to reach the site of administration [10]. To date, miltefosine (MTF) is the only oral drug approved for VL [11]. Despite its high efficacy, MTF presents problems of teratogenicity, which precludes its use in pregnant women [12]. Because of its long half-life, women receiving this treatment should take contraceptive measures for at least three months after use to reduce the risk of inducing malformations in their progeny [13]. In addition, some studies have shown the loss of efficacy of MTF in India [14] and Nepal after several years of use [15,16]. Finally, the aminoglycoside antibiotic paromomycin (PMM) is relatively ineffective as monotherapy, but is synergistically combined with other drugs, such as Sb^V^ [17] and MTF [18]. For these reasons, several international organizations such as WHO and DNDi, among others, are actively promoting research into new treatments to urgently eradicate these diseases, which represent a serious handicap to the economic and health development of endemic countries [19,20].

Among the strategies suggested for the identification of new antileishmanial compounds, phenotypic screening [21,22] of repurposing drug libraries is one of the most promising. This is due to the fact that by knowing in advance the essential safety and bioavailability parameters of the drugs, the time that must elapse from their identification to their registration can be significantly reduced [23,24]. Recently, our group screened two commercial collections of anti-infective agents, which included 1769 repurposing drugs. For this purpose, a phenotypic platform consisting of splenic explants from Balb/c mice infected with an infrared fluorescent *L. donovani* strain [25] allowed us to identify 43 compounds with antileishmanial activity at <1 μM final concentration [26]. Among the selected hits, several compounds with a nitroheterocyclic structure were identified as the most potent, including nifurtimox, nitrofurantoin, nitrofurazone, PA-824, fexinidazole and, surprisingly, the synthetic nitrofurantoin derivate Nifuratel (NFT), the latter being marketed as the active ingredient in several treatments against most agents that cause genital urinary tract infections [27,28]. NFT administered orally to Balb/c mice infected with *L. donovani* was shown to be effective, since it reduced the parasite load by more than 80% in a model of VL [26]. The mechanism of action of NFT is not well understood. It is suspected that the nitro moiety is reduced by nitroreductase 1 (NTR-1) to a nitro anion radical, which may produce toxic effects on the parasite [29]. However, other nitroheterocyclic compounds are activated by parasite nitroreductase 2 (NTR-2) [30], and the inhibition of trypanothione reductase (TR) has been identified as a possible target for this family of molecules [31]. The safety of NFT has been proven during its commercial history, which points to this compound as a promising new medicine against VL, alone or in combination with a classical antileishmanial drug.

The combination of drugs, with different mechanisms of action and synergistic effects, has been proposed by DNDi as a valid strategy for the treatment of *L. donovani* VL [32]. This strategy allows us to reduce the toxicity of the drugs involved in the combination by decreasing the concentration required to achieve the desired effect. Drug combinations reduce the generation of drug resistance as a result of the different mechanisms of action involved [33]. The combination of Glucantime with PMM is currently recommended by DNDi in East Africa [17] for the treatment of VL, and combinations of AmBisome with MTF, AmBisome with PMM, and MTF with PMM are recommended for VL in the Indian subcontinent [18,34]. Recently, the combination of MTF with PMM has been successfully applied in Eastern Africa [35].

In the present work, we have tested the combination of NFT and MTF in vitro against axenic *L. donovani* amastigotes and ex vivo against intramacrophagic parasites from mouse explants, in an attempt to find synergistic combinations able to reduce the side effects associated with the two drugs already used for the treatment of leishmaniasis. In addition, drug combinations of NFT and MTF (NFT/MTF) have been tested in vivo, showing promising results.

## 2. Results

### 2.1. Effect of NFT/MTF Combinations In Vitro and Ex Vivo

The leishmanicidal effects of NFT and MTF alone and in combination (NFT/MTF) were tested against a genetically modified strain of *L. donovani* LV9 (iRFP + *L. donovani*), which constitutively expresses the iRFP protein of the bacteriophytochrome of Rhodopseudomonas palustris when viable [25]. Intracellular macrophage-resident amastigotes were obtained from primary cultures of splenic explants from Balb/c mice infected with iRFP + *L. donovani* and fresh axenic amastigotes were recovered from bone marrow cells of infected mice. The two procedures are described in the Section 4. Infrared fluorescence emitted at 700 nm by both axenic and intramacrophage amastigotes was used as the parasite viability criterion [25].

To choose the combination ratios between both drugs, dose–response curves were performed separately with NFT and MTF in order to know their EC_50_ values in both axenic and intramacrophagic amastigotes. The emitted infrared fluorescence was plotted against different concentrations of NFT or MTF, using 0.2% DMSO or water (solvents of NFT and MTF, respectively) as negative controls (100% viability) and the complete killing effect of 10 μM AMB as a positive control (0% viability). The antileishmanial effect of both drugs was non-linearly adjusted with the SigmaPlot software package, obtaining EC_50_ values of 0.02 ± 0.00 µM for NFT and 0.63 ± 0.01 µM for MTF from axenic amastigotes (Figure 1a, Table 1). However, when the efficacy of both drugs was measured in intramacrophagic amastigotes, the EC_50_ values obtained were 0.53 ± 0.05 µM for NFT and 5.60 ± 0.38 µM for MTF (Figure 1b, Table 2). As the concentration of MTF required to produce the same antileishmanial effect was between 10 to 30 times higher than that of NFT, the NFT/MTF combination ratios chosen to determine their efficacy in the two in vitro models described above were 1/10 and 1/30, respectively. The dose–response curves for these combinations are shown in Figure 1.

The synergistic, additive or antagonistic effect of the different NFT/MTF combinations was studied following the Chow-Talalay method [36] using the “CalcuSyn” software, where the value of the “combination index” (CI) represents the type of interaction between the two drugs. This parameter is calculated from the dose–response curves fitted with the software, which includes the parameters Dm (mean dose of effect inhibiting the cells under study by 50%), m (coefficient indicating the shape of the dose–response curve) and r (correlation coefficient indicating the conformity of the data with the fitted curve) [37].

Figure 2 shows the CI values obtained for the two 1/10 and 1/30 NFT/MTF combinations, tested on axenic iRFP + *L. donovani* amastigotes, vs. the affected fraction (fa) (percentage of parasite mortality produced by the combination) after adjustment by CalcuSyn. It should be noted that CI values >1 imply an antagonistic effect of the drugs, CI~1 is an additive effect, and CI < 1 a synergistic effect. The CI values calculated for the NFT/MTF 1/10 combination were always higher than 1, thus resulting in antagonistic behavior (Figure 2a, Table 1). However, the 1/30 NFT/MTF combination was clearly synergistic over the entire fa range (Figure 2b, Table 1). Furthermore, the dose reduction index (DRI)—a parameter that measures how many times the dose of each drug in the combination can be reduced due to the synergistic effect between the two drugs—showed that the 1/30 NFT/MTF combination allows a DRI >2-fold in the case of NFT and a DRI >3-fold in the case of MTF (Table 3), thus suggesting an interesting decrease in both drugs and their potential toxic effects.

The 1/10 and 1/30 combinations of NFT/MTF were also tested ex vivo in mice splenocytes naturally infected with iRFP + *L. donovani* amastigotes (Figure 3). This model has the advantage of being a multicellular culture, in which all spleen cells are represented, including those of the immune system of the infected animal. In addition, since the amastigote resides inside the compartment where it develops naturally (macrophage phagolysosome), the model resembles largely in vivo conditions. In this model, the CI values show the antagonism of the 1/10 NFT/MTF combination in contrast to the strong synergism of the 1/30 NFT/MTF combination up to 50% fa (Figure 3, Table 2). The DRI values for the 1/30 combination at 25% fa were more than 1000-fold for NFT and MTF (Table 4).

The cytotoxicity of both drugs, alone and in combination, was determined using primary cultures of uninfected mouse spleens. For this purpose, the viability of spleen cells obtained from uninfected Balb/c mice and exposed to the highest concentrations of the tested combinations (0.5 μM NFT + 15 μM MTF) was evaluated, using the Alamar Blue staining method (Invitrogen) as a viability criterion. The results showed that the viability of spleen cells was not compromised by the combinations of both drugs and in addition, the cytotoxicity of MTF was significantly reduced when tested in combination with NFT (Figure 4).

### 2.2. Effect of NFT/MTF Combination In Vivo

To test the efficacy in vivo of the NFT/MTF combination, 6–8 week-old female Balb/c mice were infected by intraperitoneal injection with 1.5 × 10^9^ promastigotes of the light-emitting PpyRE9h + *L. donovani* strain. Mice were treated orally, 8 weeks after inoculation, with 10 mg/kg bw MTF, once a day in combination with NFT at 50 mg/kg bw twice a day for 10 days. Four additional groups were also added to the experiment in order to compare the results of the NFT/MTF combination: (i) 50 mg/kg NFT administered twice orally (NFT50); (ii) 10 mg/kg MTF administered orally once daily (MTF10); (iii) 40 mg/kg MTF administered orally once daily (MTF40); as positive control and (iv) infected as negative control (CTRL).

The parasite load was assessed in real time, by quantifying in an IVIS-Spectrum device the bioluminescence emitted by Balb/c mice infected with the transgenic strain PpyRE9h + *L. donovani* after the subcutaneous injection of 150 mg/kg bw of the light-emitting reagent D-luciferin. Light emission was recorded, animal by animal, three days after the last drug administration in all test groups and compared to the emission recorded at the beginning of the experiment to calculate the reduction in light emission produced by the different drug treatments (Figure 5a). The maximum light reduction was found in the MTF40 group (positive control), followed by the NFT/MTF combination and the MTF10 group (Figure 5b). The therapeutic effect of NFT50 was much less and more variable than that observed in the other groups and barely reduced light emission by 50% (Figure 5b).

Three days after the end of each treatment, all groups of animals were euthanized and the spleen, liver, thymus and bone marrow were dissected to evaluate the parasite load by the Limit Dilution Assay method (LDA) [39]. Figure 6 shows the reduction in parasite load of the different groups with respect to the CTRL group. In the group of animals with MTF40 (positive control), the reduction in the parasite load was almost 100% in the four samples examined (Figure 6a–d), thus demonstrating, once again, the great antileishmanial efficacy of MTF in vivo. On the other hand, the group of animals treated with MTF10 showed high percentages of reduction in the parasite load in the spleen, thymus [40] and bone marrow, but not in the liver (Figure 6a–d). The NFT50 treatment was effective in preventing parasite development in the thymus and bone marrow cells, but not in the spleen and liver, where results were similar to the CTRL group (Figure 6a–d).

Finally, the efficacy of the NFT/MTF combination was high in thymus and bone marrow cells, with similar values to those observed in the MTF40 and MTF10 groups (Figure 6c,d). In contrast, the NFT/MTF combination was much less effective in reducing the parasite load in the spleen (Figure 6b). Regarding the values obtained in the liver, the NFT/MTF-treated group showed a higher reduction (although non-significant) in parasite load than animals treated with MTF10 (Figure 6a).

### 2.3. Inhibition of TR Activity

Several studies have reported that nitrofurans act as reversible inhibitors of TR [41,42], a key enzyme involved in maintaining the redox balance of trypanosomatids by catalyzing the NADPH-dependent reduction of trypanothion. In view of these results, we decided to assess the effect of NFT on TR of *L. donovani*. Figure 7a shows the time-dependent inhibitory effect of different concentrations of NFT at saturating concentrations of oxidized trypanothione (T[S]_2_) and NADPH (0.15 mM and 0.20 mM, respectively).

Assays varying the concentrations of T[S]_2_ (from 25 μM to 150 μM) at different concentrations of NFT (from 3 μM to 15 μM) were performed. The Lineweaver-Burk double reciprocal plot of the data obtained showed uncompetitive inhibition pattern (Figure 7b) with a calculated Ki of 3.15 ± 0.50 μM.

## 3. Discussion

Therapeutic failure associated with VL is a growing problem in endemic countries, drug resistance being one of its main causes. One of the strategies to minimize the occurrence of this phenomenon is the use of therapies that combine drugs with different mechanisms of action. The drug combination is being successfully used to circumvent the resistance observed by antimonials in the Indian subcontinent and East Africa [43]. In addition, drug combination reduces the toxicity associated with the different drugs, since their synergistic effect makes it possible to reduce the doses required to achieve therapeutic efficacy. Currently, as has been done with other infectious diseases such as malaria [44], AIDS [45] or tuberculosis [46], drug combinations are being tested for the control of leishmaniasis, and some of them have been included in the WHO recommendations for the treatment of VL, such as liposomal AMB combined with PMM or MTF and antimonials combined with PMM [3].

Considering these facts, the use of drug combinations along with the repurposing strategy was considered a useful procedure in the necessary search for new, short, safe and cheap treatments for leishmaniasis. Since MTF is the only oral drug available against leishmaniasis, combinations of other drugs with this compound represent a relevant approach to achieving treatments that, with high rates of adherence, could reduce the toxic effects of the current treatments as well as the emergence of resistance. In this context, in vivo results combining the antiretroviral drug Lopinavir with MTF allowed a reduction in each compound concentration to achieve the same outcome, thereby opening the door to this combination treatment in co-infections of Leishmania with HIV [47]. More striking was the synergistic effect observed in vitro with another antiretroviral, Nelfinavir, whose leishmancidal effects were potentiated in combination with MTF [48]. Oral combination of the essential oil from *Mitracarpus frigidus* eugenol oleate with MTF showed significant antileishmanial effects compared to the individual treatment, although the type of interaction (synergy or additive effect) was not assessed [49]. AMB has also been combined with other drugs due to the emergence of resistance to this compound. In this case, in vitro assays with the antitubercular drug delamanid combined with AMB (1:1) showed a synergistic effect against intracellular *L. donovani* amastigotes [50]. Other combinations have been developed using repurposing drugs that have shown good results in vitro and in vivo. This is the case with disulfiram, which exhibited strong synergistic antileishmanial effects in vitro against *Leishmania* promastigotes in combination with amoxicillin and kanamycin [51]. Ternary combination therapies have also been implemented for the treatment of VL, including the use of the antifungal itraconazole, the hypocholesterolemic compound ezetimibe and MTF. In vivo tests in a murine model showed promising results, although the synergistic or additive effects were not [52].

In the present work, we show the strong in vitro and ex vivo synergism for NFT/MTF 1:30 combination, especially in intramacrophagic amastigotes, in the first half of the range of concentrations tested. This synergistic behavior gave rise to high DRI for both drugs used in combination. Neither the mechanism of action of MTF nor that of NFT is currently clear. Some reports point out that MTF induces apoptosis of *L. donovani* amastigotes by inhibiting phosphatidylcholine and sphingomyelin biosynthesis, which leads to an increase in intracellular ceramide, the latter is responsible for the programmed cell death of the parasite [53]. Other authors claim that MTF reduces the lipid content in the *Leishmania* membrane, thus limiting its proliferation [54]. In addition, MTF also exerts a strong pro-inflammatory response in the host, by enhancing the synthesis of inducible nitric oxide synthase 2 (iNOS2), nitric oxide (NO) and IFN-gamma, which contributes the parasite death [55]. Regarding NFT, the nitro-moiety plays a key role in its activation and toxicity. Nitrofurans like nifurtimox—as many other nitroheterocycles compounds—are enzymatically activated by type 1 nitro reductase (NTR1), which is responsible for toxic concerns about this family of compounds; bicyclic nitroheterocycles are activated by nitroreductase 2 (NTR2) [30,56]. In addition, nitrofurans were proven to be TR inhibitors. TR is the key enzyme responsible for T[S]_2_ reduction to TSH_2_, a key metabolite in controlling ROS in trypanosomatids [57]. However, the uncompetitive type of inhibition of NFT led us to think that the drug may be a “subversive” substrate rather than a true TR inhibitor [42,58]. This denomination was coined to explain the ability of nitrofurans to be reduced by the enzyme in the presence of oxygen, thereby inhibiting the reduction of T[S]_2_, consuming NADPH and generating ROS.

To define the in vivo oral doses of the NFT/MTF combination, we used the empirical data obtained in a previous study carried out in Balb/c mice in our laboratory, in which the dose of 50 mg/kg/d bid/10 days produced a significant reduction in the total parasite load and in the thymus, liver and spleen [26]. In addition, individual pharmacokinetic results obtained in previous studies for NFT and MTF in mice and humans after oral administration under similar conditions, were helpful to know the plasma concentrations of both drugs. These studies demonstrated the low bioavailability of NFT after oral administration of 50 mg/kg in mice, reaching plasma concentrations of 8 to 15 μg/mL (equivalent to 28 to 53 nM), which would be in the order of effective concentration in *Leishmania* amastigotes in vitro obtained in our studies [59,60]. Moreover, the estimated half-life of NFT was barely 3 h, thus indicating that the compound had practically been eliminated from plasma after 11 h. For this reason, we opted to use a 12 h dosing regimen during the 10 days of the experiment. This pattern proved to be safe throughout the trial as previously described [26]. With respect to MTF, snapshot pharmacokinetics were performed at doses of 3 mg/kg and 5 mg/kg, choosing 10 mg/kg bw once a day during the 10 days of the trial (1/4 of the dose used with MTF40, which served as positive control). The previous results obtained with MTF in mice demonstrated its high oral bioavailability, but only at doses higher than 30 mg/kg/d for 5 days was it able to eliminate the parasite load by >95%, even though the plasma values reached (ca. 20 μM) [61], which were already high enough for the elimination of the amastigotes, according to our studies. We decided to use the dose of 10 mg/kg/d for 10 days, which according to the previous study [61], only produced the estimated 70% clearance during the 5 days of administration. The results obtained in vivo with NFT/MTF combination showed a significant reduction in the luminescence emitted by the infected mice regarding the untreated animals and NFT50 but close to that found in the MTF10 group. The remaining luminescence coming from the spleen was visible, thus pointing to this organ as a niche barely reachable by the drugs. NFT/MTF combination reduced parasite load significantly regarding the control and NFT50 and MTF10, and was as effective as MTF40 in removing parasites from the thymus and bone marrow.

MTF is an easy-to-use oral drug with weaker side effects (nausea, vomiting, stomach pain, loss of appetite, diarrhea, headache, dizziness, drowsiness or itching) than other antileishmanial drugs [11]. However, the reproductive concerns associated with MTF administration to pregnant women, fully justify combination therapies to minimize its dosage [62]. Several studies have demonstrated embryofetal toxicity and teratogenicity in rats and rabbits administered MTF orally during organogenesis at doses much lower than the maximum recommended human dose for the body surface area [63]. These results concluded the recommendation that post-kala-azar dermal leishmaniasis patients of childbearing age should take contraceptive measures during MTF treatment and for 5 months post-treatment to prevent possible congenital anomalies in the fetus [64]. In addition to teratogenicity, reduced fertility is observed in male LV patients treated with MTF, who reported reduced ejaculation volume [65]. Unlike MTF, NFT can be used in pregnant women without the risk of teratogenicity issues. Furthermore, after more than 30 years of using NFT in clinical practice, no resistance phenomena have been described. It could be assumed that the emergence of resistances, as it has occurred with MTF after barely a decade of use, is unlikely [14].

In conclusion, NFT/MTF at 1/30 ratio has a synergic antileishanial effect both in vitro in axenic amastigotes of *L. donovani* and ex vivo in intramacrophagic forms. However, despite NFT/MTF combination produces a significant reduction in total parasite burden, especially in the thymus, liver and bone marrow, it is not able to completely clear the presence of parasites in the spleen, which behaves as a final niche of parasite accumulation difficult to reach by for this drug combination.

## 4. Materials and Methods

### 4.1. Drugs

The standard antileishmanial drug MTF (purchased from Across Organics, Fisher Scientific, Inc.) was dissolved in sterile H_2_O to a final concentration of 50 mM. NFT (purchased from MedChemExpress, NJ, USA) was dissolved to the stock concentration of 30 mM in DMSO (Sigma).

### 4.2. Experimental Animals and Ethical Statement

Experimental infections to obtain primary axenic bone marrow amastigotes and to obtain splenic explants harboring *L. donovani* amastigotes were carried out in 6–8 week-old female Balb/c mice. Animals were purchased from Janvier Laboratories (St Berthevin Cedex, France) and were received and maintained in the animal house at the University of León under standard housing conditions and with free access to food and water. The animal handling protocols used in this study comply with Spanish Law (RD 118/2021, which modifies RD 53/2013) inspired by European Union Legislation (EU 2019/1010), and were approved by the Junta de Castilla y León under authorization numbers JMJ/bb and 2113.2.

### 4.3. Parasites

The *Leishmania* strain used for in vitro and ex vivo studies was derived from *L. donovani* LV9, which was genetically modified in-house to constitutively produce the infrared fluorescent protein (iRFP) [25,66]: iRFP-*L. donovani* for the detection of viable parasites. For in vivo studies, we also used *L. donovani* LV9 strain but, this time, transfected with the firefly red-shifted luciferase PpyRE9h (PpyRE9h+ *L. donovani*) to evaluate light emission from infected organs [38]. Both strains were routinely grown as promastigotes in Schneider’s insect medium (Sigma-Aldrich, Merck, Darmstadt, Germany) supplemented with 20% (*v/v*) fetal calf serum (FBS) and an antibiotic cocktail (100 U/mL penicillin and 100 µg/mL streptomycin) at 26 °C until the mice were infected.

### 4.4. Experimental Infections and Set Up of Primary Cultures

Six- to eight-week-old female Balb/c mice were inoculated intraperitoneally with 1.5 × 10^9^ iRFP-*L. donovani* metacyclic promastigotes. After 8 to 10 weeks post-inoculation, mice were humanely sacrificed and the femur and tibia of both paws and infected spleens were aseptically dissected. To prepare a primary culture of iRFP-*L. donovani* axenic amastigotes, femur and tibia were carefully cut at both ends, and marrow cells were removed by passing warm PBS through the medullary cavity with a 29G needle. The bone marrow cell suspension was passed through a 100 µm cell strainer and centrifuged at 3500× *g* for 10 min at room temperature. After centrifugation, cells were resuspended in amastigote culture medium consisting of 15 mM KCl; 136 mM KH_2_PO_4_; 10 mM K_2_HPO_4_.3H_2_O; 0.5 mM MgSO_4_.7H_2_O; 24 mM NaHCO_3_; 22 mM glucose; 1 mM glutamine, 1× RPMI 1640 vitamin mix (Sigma-Aldrich, Merck, Darmstadt, Germany), 10 mM folic acid, 100 mM adenosine, 1× RPMI amino acid mix (Sigma-Aldrich, Merck, Darmstadt, Germany), 5 mg/mL hemin, antibiotic cocktail, 25 mM MES and 20% FBS, and incubated at 36 °C, to obtain free amastigotes [67]. To obtain intramacrophage amastigotes, spleens were cut into small pieces and were incubated with 5 mL of 2 mg/mL collagenase D (Merck, Darmstadt, Germany) prepared in buffer (10 mM HEPES, pH 7.4, 150 mM NaCl, 5 mM KCl, 1 mM MgCl_2_ and 1.8 mM CaCl_2_) for 25 min. The cell suspension was passed through a 100 µm cell strainer. After erythrocyte lysis, splenocytes were centrifuged three times at 500× *g* for 7 min at 4 °C, with PBS washes between each centrifugation. Finally, the splenocyte suspension containing intramacrophage amastigotes was resuspended in RPMI medium (Gibco, Fisher Scientific, Madrid, Spain) supplemented with 20% FBS, 1 mM sodium pyruvate, 24 mM NaHCO_3_, 2 mM L-glutamine, 1× RPMI vitamins, 25 mM HEPES and antibiotic cocktail. 

### 4.5. Axenic and Intramacrophagic Amastigotes Viability Assays

To assess the antileishmanial effect of drugs and drug combinations on axenic amastigotes, 40 μL of a suspension containing 3–10^4^ iRFP-*L. donovani* parasites per well were incubated with another 40 μL of serial dilutions (one-half or one-third dilutions) of the drug or drug combinations in the amastigote culture medium. Incubations were performed in 384-well black microtiter plates with an optical bottom at 26 °C for 72 h. A 0.1% (*v*/*v*) DMSO and 10 μM AMB solution were used as negative and positive controls, respectively [68]. 

For the ex vivo intramacrophage amastigote assay, 40 μL of one-half or one-third serial dilutions of each drug or drug combinations diluted in supplemented RPMI medium, were added to black 384-well optical-bottom black plates containing 40 μL of murine splenocytes naturally infected with iRFP-*L. donovani* (see above). Similarly, positive controls (containing 10 μM of AMB) and negative controls (containing 0.1% *v/v* DMSO), were included in all assay plates. The plates were incubated at 37 °C and 5% CO_2_, for a maximum period of 72 h [25]. 

The viability of axenic and intramacrophage amastigotes was quantified by the fluorescence emitted at 700 nm by the iRFP protein produced by live parasites and recorded by an Odyssey infrared imaging system (Li-Cor, NE, USA). The fluorescence emitted by the negative control wells—containing 0.1% (*v/v*) DMSO—was adjusted to 100% viability, whereas the fluorescence emitted by the positive control wells, containing 10 μM AMB, was adjusted to 0% viability.

To calculate the EC_50_ values for each drug or drug combination, data obtained from the measurement of fluorescence emitted by axenic and intramacrophage amastigotes were plotted against each drug concentration using the nonlinear fit analysis provided by the Sigma Plot 10.1 statistical package.

### 4.6. Cell Cytotoxicity

In order to assess the in vitro safety of the drugs and drug combinations, cytotoxicity assays were carried out in ex vivo splenic explants from uninfected mice, which were prepared as previously described [69]. Cells were microscopically counted and seeded at a density of 1 × 10^6^ cells per well in 96-well plates containing the prediluted compounds, and incubated at 37 °C and 5% CO_2_. After a 96 h incubation, Alamar Blue assay (Invitrogen, Fisher Scientific, Inc.) was used to estimate cell viability according to the manufacturer’s recommendations.

### 4.7. In Vivo Efficacy of Drug Combinations against L. donovani VL Mouse Model

To evaluate the in vivo efficacy of NFT/MTF combinations, a murine VL model, previously described by us, was used. For this purpose, 1.5 × 10^9^ promastigotes of PpyRE9h + *L. donovani* were inoculated into 6–8-week-old female Balb/c mice by intraperitoneal injection. Once the infection was established 8 weeks after inoculation, five treatment groups were established with five animals each with exception of the group treated with 40 mg/kg MTF, which included only three animals due to the availability of previous efficacy results [26,38].

The concentrations of NFT and MTF that were chosen to be tested in the NFT/MTF combination were 50 mg/kg bw twice daily for NFT and 10 mg/kg bw once daily for MTF for 10 days. Both drugs were administered orally; NFT was diluted in 0.75% carboxymethylcellulose (Merck, Darmstadt, Germany), and MTF was diluted in water. The effect of both drugs was also tested separately; thus, the final disposition of the groups was as follows: (i) negative control group (CTRL, *n* = 5) administered orally with the vehicle (carboxymethylcellulose) once daily for 10 days; (ii) nifuratel group (NFT50, *n* = 5), treated orally with NFT at 50 mg/kg bw twice daily for 10 days; (iii) miltefosine group (MTF10, *n* = 5), treated orally with MTF at 10 mg/kg bw once daily for 10 days; (iv) combination group (NFT50/MTF10, *n* = 5), treated orally with NFT at 50 mg/kg bw twice daily and MTF at 10 mg/kg bw once daily for 10 days; and (v) positive control group (MTF40, *n* = 3), treated orally with MTF at 40 mg/kg bw once daily for 5 days.

The parasite load of the animals was monitored in real time by light emission from organs infected with PpyRE9h + *L. donovani* on an IVIS Spectrum image recording system (PerkinElmer, Walthan, MA, USA). For this purpose, animals were injected subcutaneously with 150 mg/kg bw D-luciferin (PerkinElmer, Walthan, MA, USA) three days after the last drug administration. To record images, Balb/c mice were anesthetized with 2.5% (*v/v*) isofluorane (Perkin Elmer, Walthan, MA, USA) in oxygen, subsequently reduced to 1.5% and placed in the IVIS Spectrum system (PerkinElmer, Walthan, MA, USA) prewarmed to 36 °C. Bioluminescence images were acquired every 2 min for a total of 30 min. Living Imaged software (PerkinElmer, Walthan, MA, USA) was used to quantify bioluminescence, expressed as total flux (photons/s), in regions of interest (RoIs) drawn around the entire animal.

Three days after the end of treatment and after quantification of bioluminescence, the mice were euthanized. The liver, spleen, thymus and bone marrow were dissected and homogenized in PBS to estimate their parasite load by the limit dilution assay (LDA), described elsewhere [39].

### 4.8. Trypanothione Reductase (TR) Assay

Enzymatic inhibition assays of NFT on Leishmania TR were carried out using cell extracts of *L. donovani*, according to a modification of a previously described method [70]. Briefly, 1 × 10^10^ axenic *L. donovani* amastigotes were washed twice in PBS and, after centrifugation at 10,000× *g* for 5 min, the pellet was resuspended in 1 mL of lysis solution (1 mM EDTA, 40 mM HEPES, 50 mM Tris HCl pH 7.5, 2% (*v/v*) Triton X-100) containing Pierce™ Protease Inhibitors Mini Tablets (ThermoFisher Scientific Inc., Waltham, MA, USA). After incubation on ice for 15 min, 0.5 mm diameter glass beads (Merck, Darmstadt, Germany) were added and, after three 15-s vortex cycles, cell extracts were obtained by centrifugation at 10,000× *g* for 5 min at 4 °C. The enzymatic reaction was carried out in 96-well plates, which included 2 μL of different concentrations of NFT diluted in DMSO, 28 μL of TR assay solution comprising 0.2 mM NADPH (Alfa Aesar, Fisher Scientific, Inc., Ward Hill, MA, USA), variable concentrations of T[S]_2_ (Bachem, Fisher Scientific, Inc., Bubendorf, Switzerland), 0.075 mM 5,5’-dithiobis(2-nitrobenzoic acid) (DTNB) (Alfa Aesar, Fisher Scientific, Inc.) and 50 mM Tris HCl pH 7.5, and 50 μL of supernatants diluted in Tris HCl pH 7.5, the latter including 0.43 μg of total protein and initiating the reaction. Several controls were included in the assay: 2.5% DMSO (negative control), the TR assay solution without T[S]_2_ (blank reaction), and 0.1 mM thioridazine (Medchem Express, Princeton, NJ, USA) in DMSO as a positive inhibition control [71]. Enzymatic activity was measured at 412 nm for a period of up to 60 min (with 5 min intervals) at 26 °C in a Varioskan Lux spectrophotometer (Thermo Scientific, Fisher Scientific, Inc.). 

### 4.9. Statistical Analysis

The type of interaction derived from the in vitro and ex vivo MTF/NFT combinations on iRFP + *L. donovani* axenic and intramacrophagic amastigotes was described by the median-effect/combination index (CI)-isobologram equation [36,37]. The isobologram analysis involves plotting the dose–response curves for each compound and its combinations in multiple concentrations. The CI <1, =1, and >1 indicates the synergistic, additive and antagonistic effects of the drug combination, respectively. The type of interaction produced by the drug combinations was analyzed using CalcuSyn software version 2.1. (Biosoft, Cambridge, UK, 1996–2007).

The statistical evaluation of groups was done by one-way analysis of variance (ANOVA) followed by Tukey’s multiple-comparison. Differences were considered statistically significant at *p*-value < 0.05.

## Figures and Tables

**Figure 1 ijms-24-01635-f001:**
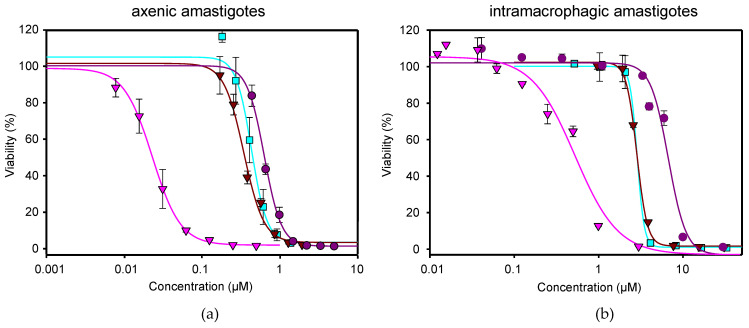
Antileishmanial effect of MTF (
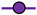
), NFT (
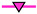
), NFT/MTF 1/10 combination (
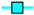
) and NFT/MTF 1/30 combination (
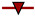
) tested in axenic amastigotes (**a**) or in intramacrophagic amastigotes (**b**). For NFT, 1/2 serial dilutions were prepared, starting with 0.5 µM for axenic amastigotes and with 3 μM for intramacrophagic amastigotes, whereas for MTF, 1/3 and 2/3 serial dilutions were performed, starting with 5 µM for axenic amastigotes and with 30 μM for intramacrophagic amastigotes. The starting concentrations in the NFT/MTF 1/10 combination were 0.125 µM/1.25 µM, with 2/3 serial dilutions for axenic amastigotes, and 3 µM/30 µM, with 1/2 serial dilutions for intramacrophagic amastigotes. In the NFT/MTF 1/30, the starting concentrations were 0.0625 µM/1.875 µM, with 2/3 dilutions for axenic amastigotes, and 0.3 µM/30 µM, with 1/2 serial dilutions. Dose–response curves were adjusted with SigmaPlot software. Results represent the mean values ± SD of three independent experiments with four technical replicates each.

**Figure 2 ijms-24-01635-f002:**
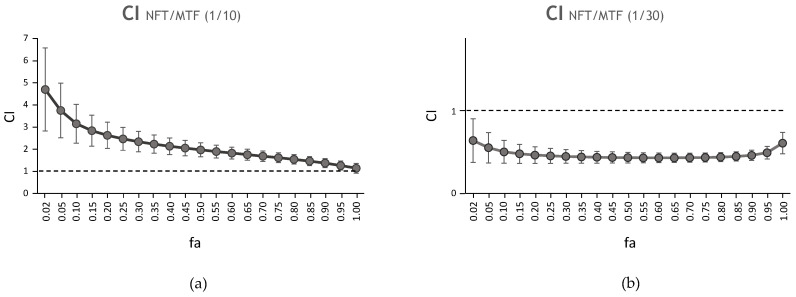
Interaction effect of the combination NFT/MTF in axenic amastigotes. (**a**) Antagonistic effect of the NFT/MTF 1/10 combination and (**b**) synergistic effect of NFT/MTF 1/30 combination, represented as CI vs. fa tested on iRFP + *L. donovani* axenic amastigotes and obtained with Calcusyn software. The data provided by CalcuSyn represent the mean of three different experiments with four technical replicates each.

**Figure 3 ijms-24-01635-f003:**
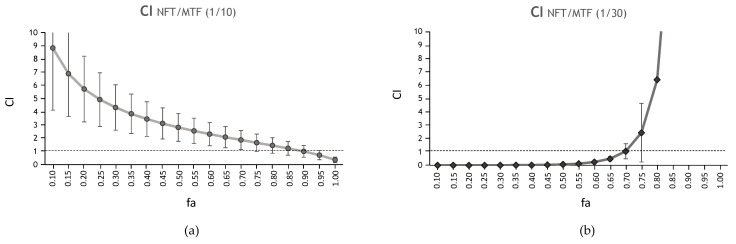
Interaction effect of the combination NFT/MTF in intramacrophagic amastigotes. (**a**) Antagonistic effect of the NFT/MTF 1/10 combination and (**b**) synergistic effect of NFT/MTF 1/30 combination, represented as CI vs. fa tested on iRFP + *L. donovani* intramacrophagic amastigotes and obtained with CalcuSyn software. The data provided by CalcuSyn represent the mean of three different experiments with four technical replicates each.

**Figure 4 ijms-24-01635-f004:**
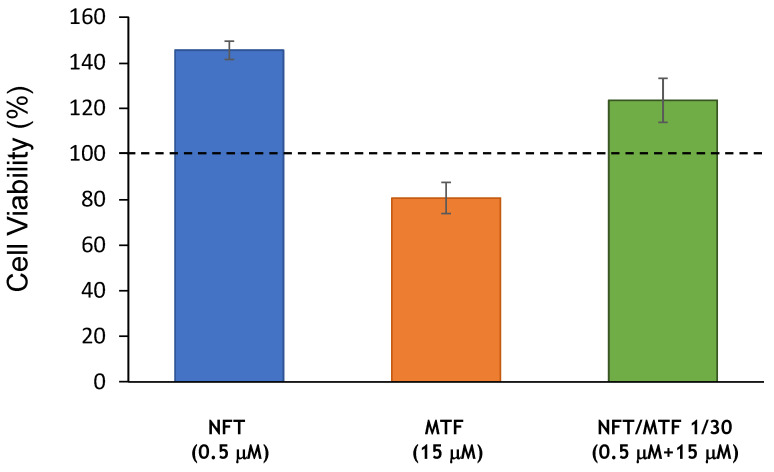
Cell viability of primary cultures of uninfected spleens in the presence of NFT, MTF, and NFT/MTF 1/30 combination.

**Figure 5 ijms-24-01635-f005:**
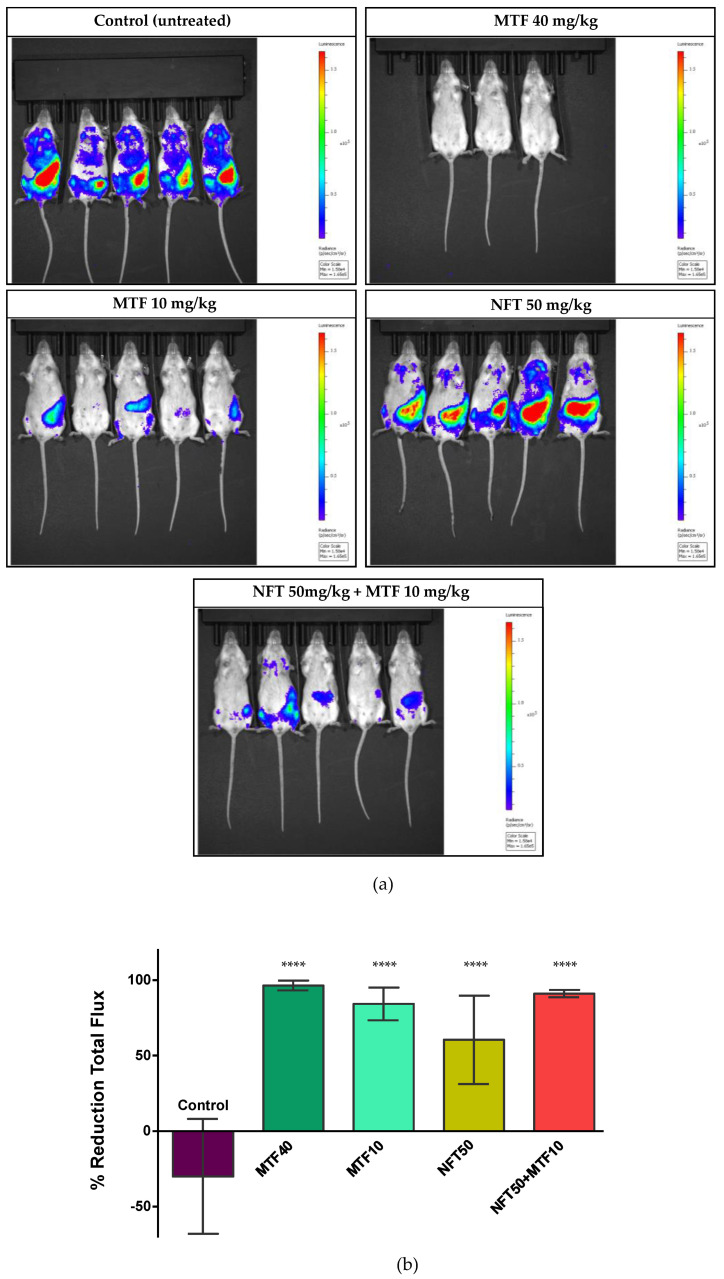
In vivo effect of the NFT/MTF combination. (**a**) Ventral images of Balb/c mice infected with PpyRE9h + *L. donovani* taken 3 days after the end of the treatment with MTF (10 mg/kg and 40 mg/kg), NFT (50 mg/kg) and the combination NFT/MTF (50 mg/kg + 10 mg/kg). Mice in the negative control group (CTRL) remained untreated. (**b**) Percentage of bioluminescence reduction. The bioluminescence emitted 3 days after the end of the treatment by the group administered with 40 mg/kg MTF (MTF40), 10 mg/kg MTF (MTF10), 50 mg/kg NFT (NFT50) and with the combination 50 mg/kg NFT + 10 mg/kg MTF (NFT50 + MTF10) was compared to the emission of bioluminescence of each group at the beginning of the experiment. Each point represents the mean of total flux reduction ± SD of the five animals of each group, with exception of the group treated with 40 mg/kg MTF, which included only three animals due to the availability of previous efficacy results [26,38]. Statistical significance (represented above each column) was calculated by one-way ANOVA (**** *p* < 0.0001).

**Figure 6 ijms-24-01635-f006:**
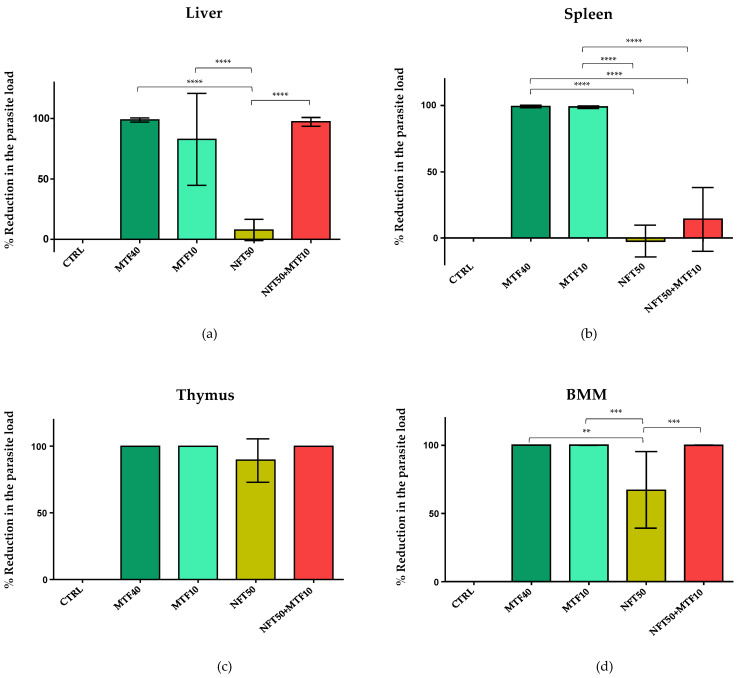
Parasite load reduction calculated by the Limit Dilution Assay (LDA) in liver (**a**), spleen (**b**), thymus (**c**) and bone marrow (BMM) cells (**d**) in the control group (CTRL, untreated), and in the different groups of animals after the treatment with 40 mg/kg MTF (MTF40), 10 mg/kg MTF (MTF10), 50 mg/kg NFT (NFT50) and with the combination 50 mg/kg NFT + 10 mg/kg MTF (NFT50 + MTF10). Each point represents the mean ± SD of the animals of each group. Statistical significance of the differences existing among groups is represented above each column, and was calculated by one-way ANOVA (** *p* < 0.01; *** *p* < 0.001; **** *p* < 0.0001). Note that in the thymus, there were no significant differences among those groups receiving treatment in the reduction in the parasite load.

**Figure 7 ijms-24-01635-f007:**
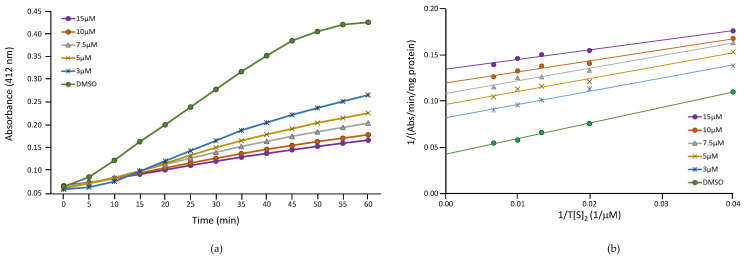
Inhibition of TR by NFT. (**a**) Time-dependent inhibitory effect of NFT on TR. The assay was performed in the presence of different concentrations of NFT (from 3 μM to 15 μM), and at 0.15 mM T[S]_2_ and 0.20 mM NADPH. (**b**) Lineweaver-Burk double reciprocal plot resulting from varying the concentrations of T[S]_2_ (from 25 μM to 150 μM) at different concentrations of NFT (from 3 μM to 15 μM).

**Table 1 ijms-24-01635-t001:** Efficacy values in axenic amastigotes and synergistic, additive or antagonistic effect of NFT, MTF and NFT/MTF combinations. EC_50_ ± SD values corresponding to 50% growth inhibition of amastigotes obtained from bone marrow cells were calculated fitting the dose–response curves with the SigmaPlot software. Dm, m, r and CI parameters for the NFT/MTF combinations were obtained using the CalcuSyn software.

					CI Values at Following Effect Levels			
Drug/s	EC_50_	Dm	m	r	25%	50%	75%	100%
NFT	0.02 ± 0.00	0.03	1.61	0.97	*N/A	*N/A	*N/A	*N/A
MTF	0.63 ± 0.01	1.74	5.29	0.96	*N/A	*N/A	*N/A	*N/A
NFT/MTF 1/10	N/A	0.05	2.72	0.99	2.46	1.96	1.61	1.36
NFT/MTF 1/30	N/A	0.01	2.40	0.97	0.57	0.52	0.49	0.50

*N/A: Not applicable.

**Table 2 ijms-24-01635-t002:** Efficacy values in intramacrophagic amastigotes and synergistic, additive or antagonistic effect of NFT and NFT/MTF combinations. EC_50_ ± SD values corresponding to 50% growth inhibition of intramacrophagic amastigotes were calculated fitting the dose–response curves with the SigmaPlot software. Dm, m, r and CI parameters for the NFT/MTF combinations were obtained using the CalcuSyn software.

					CI Values at Following Effect Levels			
Drug/s	EC_50_	Dm	m	r	25%	50%	75%	100%
NFT	0.53 ± 0.05	0.09	0.94	0.92	*N/A	*N/A	*N/A	*N/A
MTF	5.60 ± 0.38	2.19	1.20	0.95	*N/A	*N/A	*N/A	*N/A
NFT/MTF 1/10	N/A	0.36	3.34	0.94	4.93	2.82	1.65	1.00
NFT/MTF 1/30	N/A	0.002	0.23	0.84	0.002	0.06	2.43	98.06

*N/A: No applicable.

**Table 3 ijms-24-01635-t003:** Predictive drug reduction index (DRI) calculated for the NFT/MTF combinations tested in amastigotes obtained from bone marrow cells, using the CalcuSyn software.

	DRI Values at Following Effect Levels			
Drug/s	25%	50%	75%	90%
	NFT MTF	NFT MTF	NFT MTF	NFT MTF
NFT/MTF 1/10	0.45 4.54	0.59 3.73	0.78 3.07	1.03 2.52
NFT/MTF 1/30	2.27 7.68	2.84 5.99	3.56 4.66	4.46 3.63

**Table 4 ijms-24-01635-t004:** Predictive drug reduction index (DRI) calculated for the NFT/MTF combinations tested in intramacrophagic amastigotes, using the CalcuSyn software.

	DRI Values at Following Effect Levels			
Drug/s	25%	50%	75%	90%
	NFT MTF	NFT MTF	NFT MTF	NFT MTF
NFT/MTF 1/10	0.49 0.35	0.72 0.70	1.05 1.42	1.54 2.87
NFT/MTF 1/30	1253.97 1301.74	36.52 29.53	1.06 0.67	0.03 0.01

## Data Availability

Not applicable.

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
