# Peer review of "Miltefosine and Nifuratel Combination: A Promising Therapy for the Treatment of Leishmania donovani Visceral Leishmaniasis"

_ijms, 2023, doi:10.3390/ijms24021635_

Round 1
Reviewer 1 Report
In this manuscript, the authors investigate the effect of combination therapy (Miltefosine and nifuratel) for the treatment of Leishmania donovani visceral leishmaniasis. The authors demonstrated in vitro against axenic L. donovani amastigotes and ex vivo against intramacrophagic parasites from mouse explants. Also, they calculated the synergistic or antagonistic effects of that combination. The study idea was good, as visceral leishmaniasis is a tropical problem that infects a huge number of populations. The references were recent, and the discussion was comprehensive. However, some drawbacks were found, as follows:
- In the introduction section:
- The authors stated that "In addition, some studies have shown the loss of efficacy of MTF in India and Nepal after several years of use [14]," but reference number 14 was about two cases in India and didn’t mention any cases in Nepal; the authors must revise this.
- In the results section:
- The authors claimed that the combination therapy would decrease the toxicity of the drugs; however, no liver or kidney function tests were analyzed in the study. The authors must analyze the biochemical parameters and, if applicable, investigate the histopathological examination of the liver, spleen, and kidney.
- In the materials and methods section:
- The animal groping is not clearly described; authors must rewrite the groups and describe each treatment dosage and administration kind (oral, IP) for each group.
Author Response
The authors stated that "In addition, some studies have shown the loss of efficacy of MTF in India and Nepal after several years of use [14]," but reference number 14 was about two cases in India and didn’t mention any cases in Nepal; the authors must revise this.
Thank you for this observation. It is true that the article cites in the introduction the cases of MTF resistances in references 9, 10 and 23, although it does not specifically describe more than Indian cases. For this reason we have introduced references 15 and 16 in the new MS (lines 56 and 57, respectively).
15.- Rijal S, Ostyn B, Uranw S, Rai K, Bhattarai NR, Dorlo TPC, et al. Increasing failure of miltefosine in the treatment of Kala-azar in Nepal and the potential role of parasite drug resistance, reinfection, or noncompliance. Clin Infect Dis. 2013;56:1530–8.
16.- Shaw CD, Lonchamp J, Downing T, Imamura H, Freeman TM, Cotton JA, Sanders M, Blackburn G, Dujardin JC, Rijal S, Khanal B, Illingworth CJ, Coombs GH, Carter KC. In vitro selection of miltefosine resistance in promastigotes of Leishmania donovani from Nepal: genomic and metabolomic characterization. Mol Microbiol. 2016 Mar;99(6):1134-48.
The authors claimed that the combination therapy would decrease the toxicity of the drugs; however, no liver or kidney function tests were analyzed in the study. The authors must analyze the biochemical parameters and, if applicable, investigate the histopathological examination of the liver, spleen, and kidney.
We appreciate the referee's question, but the aim of this study was to demonstrate the synergistic effect of two clinically used drugs in combination on LV both in vitro, ex vivo and in vivo, based on the existing literature data on the safety of both compounds when applying a drug repurposing approach.
The reason for stating that the combination therapy would decrease the toxicity of the drugs is based on the fact that in the ex vivo combination trials we demonstrated a very significant DRI (drug reduction index) (Tables 3 and 4 of the MS), which allows us to make in vivo combinations of at least miltefosine (drug with higher bioavailability in vivo).
Secondly, the ex vivo cytotoxicity data of the drug combination are equally very promising at maximal concentrations of both drugs (Fig. 4 of the current MS).
Thirdly we are using two repurposing drugs whose low in vivo liver toxicity is well known (two texts with their references about the liver toxicity of both compounds are included).
NFT toxicity: "Nifuratel proved to be practically non-toxic in acute tests in mice and rats. Indeed, it was impossible to determine the LD50, since the single doses up to 5 g/kg orally and up to 2 g/kg intraperitoneally resulted in no mortality even after a 5-day oral administration to mice and rats" [(Arzneim.-Forsch./Drug Res. 52, No. 1, 8-13 (2002)].
MTF toxicity: "There have been no case reports of clinically apparent liver injury with jaundice attributed to miltefosine therapy. Thus, significant liver injury from miltefosine must be very rare, if it occurs at all. Likelihood score: E (unlikely cause of clinically apparent liver injury)." [LiverTox: Clinical and Research Information on Drug-Induced Liver Injury [Internet]. Bethesda (MD): National Institute of Diabetes and Digestive and Kidney Diseases; 2012-. Miltefosine].
Fourthly, we have shown in this study the synergistic effect on the LV model. Further preclinical studies will analyze the pharmacokinetic and pharmacodynamic profile of the combination as well as possible harmful effects, if any of the combination.
The animal grouping is not clearly described; authors must rewrite the groups and describe each treatment dosage and administration kind (oral, IP) for each group.
Thank you for the comment. We have rewritten paragraph 4.7. (In vivo efficacy of drug combinations against L. donovani VL mouse model) in the materials and methods section, and included detailed information about animal grouping. The new paragraph is as follows:
The concentrations of NFT and MTF chosen to be tested in the NFT/MTF combination were 50 mg/kg bw twice daily for NFT and 10 mg/kg bw once daily for MTF for 10 days. Both drugs were administered orally; NFT was diluted in 0.75% carboxymethyl-cellulose (Merck, Darmstadt, Germany), and MTF was diluted in water. The effect of both drugs was also tested separately; thus, the final disposition of the groups was as follows: i) negative control group (CTRL, n=5) administered orally with the vehicle (carboxymethylcellulose) once daily for 10 days; ii) nifuratel group (NFT50, n=5), treated orally with NFT at 50 mg/kg bw twice daily for 10 days; iii) miltefosine group (MTF10, n=5), treated orally with MTF at 10 mg/kg bw once daily for 10 days; iv) combination group (NFT50/MTF10, n=5), treated orally with NFT at 50 mg/kg bw twice daily and MTF at 10 mg/kg bw once daily for 10 days; and v) positive control group (MTF40, n=3), treated orally with MTF at 40 mg/kg bw once daily for 5 days.
Reviewer 2 Report
In this work, Estela Melcón-Fernández et. al. described combinations of miltefosine with nifuratel for the treatment of Leishmania donovani visceral leishmaniasis. The result demonstrated that they have a potent synergy in axenic amastigotes from bone marrow, in intracellular amastigotes from infected Balb/c mouse spleen macrophages and in vivo. This work can be accepted in its present form.
Author Response
Thanks for your kind response
Round 2
Reviewer 1 Report
I think the authors have addressed all the questions and comments.